# Essential Roles of PPARs in Lipid Metabolism during Mycobacterial Infection

**DOI:** 10.3390/ijms22147597

**Published:** 2021-07-15

**Authors:** Kazunari Tanigawa, Yuqian Luo, Akira Kawashima, Mitsuo Kiriya, Yasuhiro Nakamura, Ken Karasawa, Koichi Suzuki

**Affiliations:** 1Department of Molecular Pharmaceutics, Faculty of Pharma-Science, Teikyo University, Itabashi-ku, Tokyo 173-8605, Japan; tanigawa@pharm.teikyo-u.ac.jp (K.T.); nakayasu@pharm.teikyo-u.ac.jp (Y.N.); karasawa@pharm.teikyo-u.ac.jp (K.K.); 2Department of Clinical Laboratory Science, Faculty of Medical Technology, Teikyo University, Itabashi-ku, Tokyo 173-8605, Japan; yuqianluo31@foxmail.com (Y.L.); akirak5243@gmail.com (A.K.); mkiriya0226@med.teikyo-u.ac.jp (M.K.); 3Department of Laboratory Medicine, Nanjing Drum Tower Hospital, Nanjing University Medical School, Nanjing 210008, China

**Keywords:** mycobacteria, *M. tuberculosis*, *M. leprae*, PPARs, lipid droplets

## Abstract

The mycobacterial cell wall is composed of large amounts of lipids with varying moieties. Some mycobacteria species hijack host cells and promote lipid droplet accumulation to build the cellular environment essential for their intracellular survival. Thus, lipids are thought to be important for mycobacteria survival as well as for the invasion, parasitization, and proliferation within host cells. However, their physiological roles have not been fully elucidated. Recent studies have revealed that mycobacteria modulate the peroxisome proliferator-activated receptor (PPAR) signaling and utilize host-derived triacylglycerol (TAG) and cholesterol as both nutrient sources and evasion from the host immune system. In this review, we discuss recent findings that describe the activation of PPARs by mycobacterial infections and their role in determining the fate of bacilli by inducing lipid metabolism, anti-inflammatory function, and autophagy.

## 1. Introduction

The *Mycobacterium* genus was one of the first bacterial genera described. The most characteristic feature of mycobacteria is resistance to acid alcohol, which is utilized for Ziehl–Neelsen staining. Pathogenic mycobacteria can be categorized into three groups: *Mycobacterium tuberculosis* (*M. tuberculosis*) complex, which causes tuberculosis; *M. leprae* and *M. lepromatosis*, which both cause leprosy; and atypical mycobacteria or nontuberculous mycobacteria (NTM), which are mycobacteria responsible for a wide range of diseases. Mycobacterial cell walls consist of large amounts of lipids (30% to 40% of the total weight) that form a complex tripartite structure. The lipids are major effector molecules that affect the physiology of both the host cells and the bacilli by modulating their metabolism and stimulating immune responses to the bacilli. Most pathogenic mycobacteria, including *M. leprae*, utilize lipids from the host as a source of nutrients and to evade the immunity from the host, enabling the bacteria to both hide and replicate within host cells.

The transcription factors known as peroxisome proliferator-activated receptors (PPARs) were discovered in 1990 as enhancers of peroxisome proliferation in rodents [1] and belong to the ligand-activating nuclear hormone receptor (NR) superfamily. PPARs form heterodimers with retinoid X receptors (RXRs), enabling them to bind PPAR-responsive regulatory elements (PPRE) located in the promoter regions of their target genes. Three types of PPARs have been identified in mammals: PPAR-α (NR1C1), PPAR-β/δ (NR1C2), and PPAR-γ (NR1C3) [1,2]. Each PPAR is encoded by a separate gene and is expressed in amphibians [3], rodents [4,5], and humans [6,7]. PPAR-α and PPAR-γ are conserved proteins expressed in wide varieties of species, whereas PPAR-β/δ has diverged considerably [5]. PPARs respond to ligands and regulate the transcription of target genes. The role of PPARs is to modulate the expression of genes central to regulating glucose, lipid, and cholesterol metabolism.

It has been reported that the induction of PPAR-γ by the Middle East respiratory syndrome coronavirus (MERS-CoV) is necessary for infection. The PPAR-γ activation is mediated by the MERS-CoV-derived S glycoprotein along with concurrent inhibition of macrophage responses and the suppression of proinflammatory cytokines [8]. It has also been reported that PPAR-γ activation maintains the viral infection by inducing lipid metabolism. Monocytes isolated from coronavirus disease 2019 (COVID-19) patients show an accumulation of lipid droplets compared with other donors [9]. Infection with SARS-CoV-2 modulates lipid uptake and synthesis pathways by inducing PPAR-γ expression in monocytes; lipid droplet formation is also triggered in multiple human cell lines [9].

Recently, the lipid metabolism pathway used by mycobacteria in host cells has been revealed, and the involvement of PPARs clarified. In this review, we focus on the involvement of PPARs in host–mycobacteria crosstalk.

## 2. Activation of PPARs by Mycobacteria

PPARs are activated by endogenous and exogenous compounds. For instance, eicosanoids and long-chain fatty acids (LCFAs) are the endogenous ligands for PPAR-α and PPAR-β/δ [10,11]. PPAR-γ is activated by metabolites of arachidonic acid, such as 5-oxo-eicosatetraenoic acid (5-oxo-ETE) and 5-oxo-15(S)-hydroxyeicosatetraenoic acids (5-oxo-15(S)-HETE) [12,13], in addition to oxidized low-density lipoprotein (oxLDL) derivatives [14]. Several exogenous compounds are highly specific activators and modulators for mammalian PPAR subtypes: PPAR-α by the hypolipidemic drugs clofibrate and fenofibrate and the synthetic ligand Wy-14643 and PPAR-γ by the thiazolidinedione (TZD) group of antidiabetic drugs (including rosiglitazone, ciglitazone, troglitazone, and pioglitazone) [15]. GW501516, GW0742, and bezafibrate are highly selective PPAR-β/δ agonists, while GW1929 and GW2090 are specific PPAR-γ activators [16]. 

PPARs are also activated by mycobacterial infection; however, the bacterial component(s) responsible are not well understood. Organisms that naturally produce unsaturated fatty acids at the C10 position are relatively rare in nature, while several mycobacteria species, including *M. vaccae*, are able to accomplish this desaturation [17,18,19,20]. The mycobacteria-derived 10 (Z)-hexadecenoic acid upregulates genes in the PPAR signaling pathway and represses the proinflammatory cytokines in macrophages [21]. Furthermore, 10 (Z)-hexadecenoic acid and monoacylglycerol (MAG), which contains 10 (Z)-hexadecenoic acid, both activate PPAR-α but have no effect on PPAR-γ or PPAR-δ. The observed effects are blocked by PPAR-α antagonists and absent in PPAR-α-deficient mice. Recently, we found that PPAR-γ and PPAR-δ are activated in *M. leprae*-infected macrophages [22]. Infection with a recombinant strain of *M. bovis* BCG that produces phenolic glycolipid-1 (PGL-1) of *M. leprae* activates PPAR-γ in primary cultures of human Schwann cells [23].

Mannose-capped lipoarabinomannan (ManLAM) is present in the members of the *M. tuberculosis* complex, which interact with the mannose receptor (MR) in alveolar macrophages (AMs). High levels of PPAR-γ are expressed in activated AMs and macrophage-derived foam cells [24,25]. ManLAM upregulates PPAR-γ expression in human macrophages, consistent with *M. tuberculosis* infection. Furthermore, activation by ManLAM is suppressed by MR siRNA. These results indicate that the activation of PPAR-γ by *M. tuberculosis* is due to the interaction between its cell wall component ManLAM and host MRs.

Several molecules are known to bind to PPARs, including polyunsaturated fatty acids (PUFAs), such as certain ω3-PUFAs (e.g., docosahexaenoic acid with C22:6 and α-linolenic acid with C18:3) and certain ω6-PUFAs (e.g., arachidonic acid with C20:4 and linoleic acid with C18:2) [26,27]. Saturated fatty acids, such as stearic acid with C18:0 and myristic acid with C14:0, also bind to PPAR-α. *M. leprae* cell wall lipids also contain mycolic acids, other types of LCFAs typical for mycobacteria, such as alpha-mycolic acids and ketomycolic acids [28]. However, whether or not this lipid could be a ligand for PPARs is not known.

## 3. The Roles of PPARs in Lipid Metabolism

The function of PPARs, including PPAR-α, PPAR-β/δ, and PPAR-γ, is closely involved in lipogenesis and lipid metabolism in triacylglycerol (TAG) and cholesterol synthesis. PPAR-α plays an important role in the regulation of cholesterol and metabolism of bile acid. In the fasting state, PPAR-α accelerates fatty acid formation in the liver by regulating apolipoprotein expression. This increases high-density lipoprotein cholesterol (HDL-C) in the plasma and reduces low-density lipoprotein cholesterol (LDL-C) levels [29,30]. In addition, PPAR-α mediates cholesterol transport by enhancing the expression of the apolipoprotein AI (Apo-AI). Thus, PPAR-α stimulates the expression of the liver X receptor (LXR), which regulates ATP-binding cassette transporter A1 (ABCA1) expression, increases the production of HDL (rich in Apo-AI), and induces outflow of cholesterol from macrophages [31]. Therefore, fibrates that activate PPAR-α have a beneficial effect on reducing TAG and LDL-C (arteriosclerotic lipids), as well as increasing HDL-C levels in plasma [29,32]. 

Apolipoprotein E (ApoE)^−/−^ mice fed a high cholesterol diet have high plasma concentrations of LDL-C and develop atherosclerosis. The potential PPAR-γ agonist Danshensu Bingpian Zhi (DBZ) prevents atherosclerosis by modulating the expression of LXR to inhibit foam cell formation and inflammatory response [33,34]. PPAR-β/δ agonists seem to have similar effects as PPAR-α and PPAR-γ agonists by increasing plasma HDL levels while lowering LDL. These effects have been tested in both primate and rodent models [35,36]. In addition, PPAR-β/δ reduces the expression of Niemann-Pick C1-like 1 (NPC1L1), a cholesterol importer in the intestinal cells; reduces cholesterol absorption; and improves intestinal cholesterol outflow [37].

Intracellular TAG synthesis requires fatty acid metabolism and glucose homeostasis regulation. PPAR-α promotes glycolysis and de novo synthesis of fatty acid, while it decreases gluconeogenesis. Thus PPAR-α has an antagonistic function in glucose homeostasis to reduce lipid accumulation by suppressing glycolysis and enhance glycogen synthesis and fatty acid oxidation (FAO) [38]. These effects of PPAR-α were observed following PPAR-α overexpression in mouse skeletal muscle, which resulted in increased glucose and insulin levels in the plasma [39].

PPAR-β/δ has an important role in improving glycolysis and glucose uptake as well as glycogen storage, while suppressing gluconeogenesis [40,41]. PPAR-β/δ synergistically improves the catabolism of fatty acid and suppresses lipogenesis [42]. It has also been reported in the liver that PPAR-β/δ reduces the stability of sterol regulatory element-binding protein C (SREBP1C), which enhances lipogenesis by activating insulin-induced gene 1 (Insig-1) and preventing lipid accumulation [43]. Furthermore, PPAR-β/δ enhances the thermogenesis of brown adipose tissue (BAT) by regulating the transcription of FAO enzymes and uncoupling protein 1 (UCP-1) [44].

PPAR-γ directly binds the promoter region of various adipogenic genes, suggesting that it is an essential factor for adipogenesis [45]. Activated PPAR-γ reduces the amount of free fatty acid to increase the storage of TAG in adipose tissue [46]. PPAR-γ stimulates the differentiation of preadipocytes into adipocytes [47] and regulates the sensitivity of insulin in tissues and fatty acid storage by modulating genes involved in the release, transport, and storage of fatty acids in mature adipocytes. Furthermore, PPAR-γ transcriptionally activates the genes encoding c-Cbl-associated protein (CAP) and glucose transporter type 4 (Glut4), and contributes to glucose metabolism [48]. This evidence is consistent with our previous reports that *M. leprae* promotes the activation of PPAR-γ and increases intracellular TAG levels in THP-1 cells. Therefore, it is suggested that the activation of PPAR-γ is important for the increase of TAG and cholesterol in the formation of lipid droplets following mycobacterial infection.

## 4. Emerging Roles of PPARs in Lipid Metabolism during Mycobacteria Infection

Mycobacterial infection induces lipid droplet formation in macrophages. These lipids are essential for mycobacterial survival and are presumed to be a carbon source. In several different models, *M. tuberculosis* has been shown to use accumulated lipids as a carbon source at various stages of the infectious process [49,50,51,52]. *M. tuberculosis*-induced lipid droplets in macrophages primarily contain cholesterol esters and TAG. The cholesterol is transported through the bacterial cell membrane by Mce4, a bacterial lipid transporter required for cholesterol import and its utilization [53,54]. Many of the active compounds that limit *M. tuberculosis* growth in macrophages have been found to inhibit cholesterol-related processes, indicating that cholesterol is central to *M. tuberculosis* infection [55]. Fatty acids are also an abundant lipid in human granulomas [56]. Although it has been thought that *M. tuberculosis* assimilates and metabolizes fatty acids, recent genome sequencing has identified many putative fatty acid β-oxidation genes [57].

Since *M. leprae* has lost the *mce4* operon, *M. leprae* seems to use cholesterol oxidase (ML1492) in order to convert cholesterol to cholestenone for survival [58]. In leprosy skin tissue sections, *M. leprae*-containing histiocytes and Schwann cells are filled with cholesterol [59,60]. This has been confirmed with the observation of cholesterol accumulation in *M. leprae*-infected primary macrophage [60,61]. Furthermore, the expression of cholesterol synthase, HMG-CoA reductase, was increased following infection, and when de novo cholesterol synthesis was inhibited by lovastatin, viability of *M. leprae* was reduced [61].

Conversely, high-performance thin-layer chromatography (HPTLC) analysis demonstrates that TAG is the main component of the lipid in *M. leprae*-infected human monocytic THP-1 cells [62]. It has been reported in Schwann cells that *M. leprae* infection enhances glucose uptake and stimulates the pentose phosphate pathway, which is required for TAG synthesis [63]. The accumulated TAGs are maintained by the enhanced expression of adipose differentiation-related protein (ADRP) and perilipin and by the reduced expression of hormone-sensitive lipase (HSL), which contributes to lipid degradation [64,65]. Glycerol-3-phosphate acyltransferase 3 (GPAT3) is an important rate-limiting enzyme for TAG synthesis [66]; accordingly, the internalization and viability of bacilli are lower in *GPAT3* knockout cells [62]. Furthermore, clofazimine, a therapeutic agent for leprosy, reduces the accumulation of lipid in *M. leprae*-infected THP-1 cells and promotes the production of interferon (IFN)-β and IFN-γ [67]. Therefore, mycobacterial viability is hypothesized to be closely related to lipid metabolism in host cells, especially the accumulation of TAG and cholesterol.

A recent study demonstrated that PPAR-mediated lipid metabolism is a key process in foamy cell formation following *M. leprae* infection. Among PPARs, the involvement of PPAR-γ in mycobacterial infections has been studied. Infection with *M. tuberculosis* modulates homeostasis of host lipid and induces foamy macrophages, which is necessary for intracellular parasitization and growth [68,69]. The virulent H37Rv strain of *M. tuberculosis* induces PPAR-γ expression [25], while attenuated *M. bovis* BCG slightly upregulates PPAR-γ [25,70]. In vitro interference with PPAR-γ signaling in *M. tuberculosis*-infected macrophages decreases intracellular lipid accumulation and increases mycobacterium killing [71]. Pretreatment with a PPAR-γ antagonist significantly suppressed mycobacterial (*M. bovis* BCG and *M. tuberculosis*) induction of intracellular lipid droplet accumulation [70,71,72]. In addition, *M. tuberculosis* growth was attenuated in human lung macrophages after PPAR-γ deletion or isolation from PPAR-γ-deficient mice. Taken together, these data suggest that PPAR-γ is required for foam cell formation in tuberculous granulomas, which is related to bacilli survival.

Recently, in *M. leprae*-infected THP-1 cells, we reported that the increased expression of PPAR-γ and PPAR-δ coincided with the induction of intracellular lipid droplet formation [22]. Further, the expression of the PPAR-γ target genes *ADRP*, scavenger receptor *CD36*, fatty acid-binding protein 4 (*FABP4*), and apolipoprotein C-1 (*APOC1*) were significantly increased. Activation of the PPAR-γ signaling pathway is responsible for the upregulation of *Gpat3* expression during adipocyte differentiation [73,74,75]. We also found that GPAT3 expression is induced in THP-1 cells infected with *M. leprae*, suggesting that the mechanism of intracellular TAG accumulation is triggered by PPAR-γ activation [62].

The expression of CD36, an essential receptor for LDL-C incorporation, is also induced by *M. tuberculosis* through PPAR-γ in THP-1 macrophages [71]. CD36 can interact with surfactant in the lungs and promote the proliferation of *M. tuberculosis* in human macrophages in vitro [76]. CD36 directly interacts with TLR2 in macrophages infected with *M. bovis* BCG, as demonstrated by co-immunoprecipitation [77]. The neutralization of CD36 subsequently decreases PPAR-γ expression and lipid droplet formation and prostaglandin E2 (PGE2) secretion. In addition, *M. tuberculosis* upregulates the expression of GLUT1 and GLUT3 on the cell membrane by PPAR-γ activation of glucose metabolism. Its activation is suppressed by the PPAR-γ inhibitor T0070907 but enhanced by the agonist pioglitazone [78]. These data suggest that the activation of PPAR-γ promotes cholesterol and TAG uptake, both of which are components of the lipid droplets in mycobacteria-infected macrophages. Cholesterol accumulation in infected macrophages reduces cell wall permeability to rifampin, one of the first-line antituberculosis drugs, and masks surface antigens of mycobacteria [79]. Thus, lipids also play a role in drug resistance. 

On the other hand, PPAR-α is known to promote the metabolism of lipids accumulated in *M. tuberculosis*-infected macrophages and suppress lipid droplet formation. Following infection with *M. tuberculosis*, PPAR-α^-/-^ bone marrow-derived macrophages decrease the activation of the transcription factor EB (TFEB), a responsible factor for the regulation of autophagy, and increase lipid droplet formation. Conversely, PPAR-α activation significantly reduces the amount of lipid droplets in mycobacteria-infected macrophages, suggesting that PPAR-α promotes lipid catabolism in mycobacterial infection [80]. Thus, PPAR-α and PPAR-γ may have opposed roles in the host defense during mycobacterial infection.

## 5. Anti-Inflammatory Effects Are Mediated by PPARs during Mycobacterial Infection

PPAR-α and PPAR-γ may also have opposing functions in the immune response to mycobacterial infections. PPAR-α deficiency leads to excessive proinflammatory cytokine and chemokine production after lung and macrophage infection. The deletion of PPAR-α in mice increases the expression of interleukin (IL)-6 and tumor necrosis factor (TNF)-α as well as neutrophil recruitment following *M. tuberculosis* infection [80]. Infection of PPAR-α-deficient mice with *M. abscessus* also increased the intracellular bacterial load and histopathological damage [81].

However, PPAR-γ suppresses IL-1β, IL-6, and TNF-α in phorbol-12-myristate-13-acetate (PMA)-stimulated human monocytes [82]. Phagocytosis of *M. tuberculosis* by human macrophages activates PPAR-γ via the mannose receptor (CD206) [25], which reduces the proinflammatory response. Deletion of PPAR-γ in pulmonary macrophages enhanced proinflammatory cytokines and reduced *M. tuberculosis* growth in murine models [83]. Similarly, *M. bovis* BCG infection enhances PPAR-γ expression through TLR2 in mouse macrophages, which regulates lipid droplet formation and PGE2 production [70]. Synthetic aptamers (ZXL1) against ManLAM inhibits immunosuppression of CD11c and enhances presentation of *M. tuberculosis* antigens in dendritic cells (DCs) [84]. In this process, PPAR-γ expression is downregulated, thereby enhancing mRNA expression and cytokine production of IL-1β and IL-12 and decreasing anti-inflammatory cytokine IL-10 production in ManLAM-treated macrophages [84,85]. Therefore, the PPAR-γ expression induced by mycobacterium infection is considered important for the suppression of the immune response in the host cells.

## 6. Mycobacteria-Induced PPAR-Mediated Autophagy

Autophagy, a cytoplasmic degradation system, plays an important role in the host defense against intracellular bacteria. PPAR-α induces autophagy and ameliorates inflammatory and injurious conditions in many cell types [86,87]. PPAR-α modulates antimicrobial responses to *M. tuberculosis*, *M. bovis* BCG, and *M. abscessus* by TFEB [80,81,88]. TFEB is an important regulator of autophagy, lipid catabolism, and lysosomal function [89,90,91]. PPAR-α transactivates autophagy-related genes (ATGs) to promote autophagy [80,92]. Importantly, there is considerable evidence for crosstalk between PPAR-α and TFEB [80,89,90]. A recent study showed that sirtuin 3 (SIRT3) induced antibacterial autophagy during *M. tuberculosis* infection through PPAR-α [88], indicating that PPAR-α may function in the host defense against intracellular *M. tuberculosis* by mediating autophagy.

PPAR-γ is less studied for its involvement in autophagy during mycobacterial infection. A DNA microarray analysis of mouse macrophage J774 cells treated with ManLAM alone or with the PPAR-γ inhibitor GW9662 showed that the inhibitor downregulates the expression of the AMP-activated protein kinase (AMPK) regulatory 2 subunit *(Prkag2*), which activates AMPK [93]. AMPK, an essential regulator of autophagy [94], is required for the phagosome–lysosome fusion [95]. These data suggest a regulatory role for PPAR-γ signaling in autophagy. In addition, recent studies have revealed that either an AMPK activator (5-aminoimidazole-4-carboxamide 1-β-D-ribofuranoside, AICAR), a SIRT1 activator (e.g., resveratrol), or a SIRT3 activator (konokiol) can promote anti-mycobacterial activity through autophagy induction or AMPK activation [88,95,96,97]. It was shown in a previous study that lipid droplets are transported to lysosomes through the autophagy pathway, thus presenting the possibility that lysosomal acid lipase hydrolyzes lipid droplets [98]. Therefore, autophagy may be an essential mechanism in the regulation of lipid metabolism in macrophages during *M. tuberculosis* infection.

## 7. Conclusions

PPARs are important to the host-dependent mechanism of lipid metabolism and accumulation during mycobacterial infection. After the infection of macrophages by *M. leprae* or *M. tuberculosis*, PPAR-γ is activated and translocated into the nucleus to regulate genes that contribute to lipid metabolism, accumulation, and uptake (Figure 1). During cholesterol accumulation, APOC1 promotes the formation of LDL-C through the lipidation of very-low-density lipoprotein (VLDL) and chylomicron (CM), which is then imported by CD36. CD36 also imports fatty acids, which are transported to several intracellular organelles by FABP4. Glycerol-3-phosphate (G3P) is synthesized from glucose taken up by GLUT1/3, and GPAT3 esterifies fatty acids to promote TAG synthesis.

Since the cell wall lipids of mycobacteria are complex constructions of several lipid components, it is possible that the lipids that accumulate in the host cells are used for both immune evasion and the construction of the mycobacterial cell wall. Although many endogenous and exogenous PPAR ligands have been identified, the ligands essential for mycobacteria infection are still poorly understood. Since the host transcriptional profiles conducted after inoculation of live or dead bacteria are significantly different (unpublished observation), such an approach may be appropriate for identifying the unknown factor(s) specific in live mycobacteria. As described, lipid accumulation in the host is associated with mycobacterial survival; therefore, existing lipid metabolism inhibitors may be potential antimycobacterial agents. The PPARs that play important roles in lipid metabolism, anti-inflammatory action, and autophagy may be novel therapeutic targets in mycobacterial infections.

## Figures and Tables

**Figure 1 ijms-22-07597-f001:**
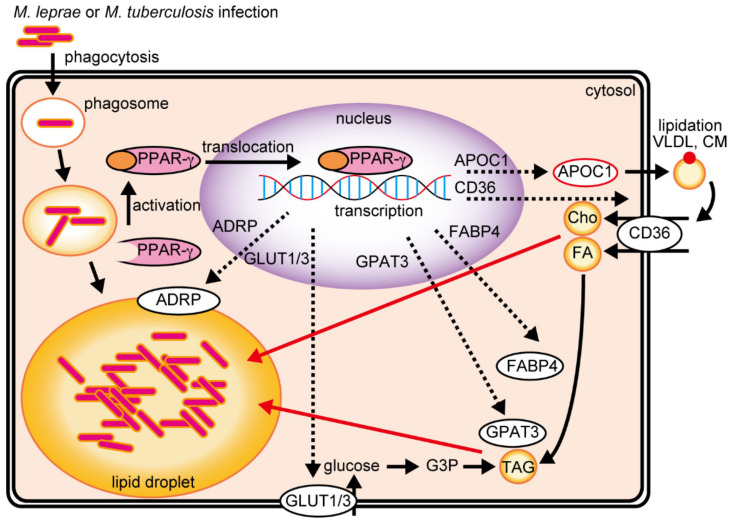
*M. leprae* and *M. tuberculosis* infections activate PPAR-γ to induce lipid droplet formation in the host cell. APOC1 binds to extracellular VLDL cholesterol or CM and is followed by LDL cholesterol uptake via CD36, thus accumulating intracellular cholesterol. Intracellular TAG accumulation is induced by two PPAR-γ-mediated pathways. FABP4 acylates extracellular fatty acids taken up by CD36 and is utilized by GPAT3 for TAG synthesis. GLUT1/3 induces intracellular glucose uptake and is subsequently utilized by GPAT3 for TAG synthesis. APOC1, apolipoprotein C-1; VLDL, very-low-density lipoprotein; CM, chylomicron; LDL, low-density lipoprotein; FABP4, fatty acid-binding protein 4; FA, fatty acid; TAG, triacylglycerol; GPAT3, glycerol-3-phosphate acyltransferase 3; GLUT1/3, glucose transport protein type 1/3.

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
