# Peer review of "Essential Roles of PPARs in Lipid Metabolism during Mycobacterial Infection"

_ijms, 2021, doi:10.3390/ijms22147597_

Round 1
Reviewer 1 Report
The authors tried to collect the latest research results of PPARs function and the the relations with mycobacteria infection, to organize them into several subtopics: Activation of PPARs by mycobacteria, The roles of PPARs in lipid metabolism, Emerging roles of PPARs in lipid metabolism during mycobacteria infection, Anti-inflammatory effects are mediated by PPARs during mycobacterial infections, and Mycobacterial-induced PPAR-mediated autophagy. I believe the authors did a great job, but something needs to be modified:
L100: "M. leprae cell wall lipids also contain an LCFA called mycolic acid."
I don't think that M. leprae cell wall contains only one kind of mycolic acid. According to "Quantitative Comparison of the Mycolic and Fatty Acid Compositions of Mycobacterium leprae and Mycobacterium gordonae", M. leprae contains alpha-mycolic acids and ketomycolic acids.
Author Response
Response to Reviewer 1 Comments
The authors tried to collect the latest research results of PPARs function and the the relations with mycobacteria infection, to organize them into several subtopics: Activation of PPARs by mycobacteria, The roles of PPARs in lipid metabolism, Emerging roles of PPARs in lipid metabolism during mycobacteria infection, Anti-inflammatory effects are mediated by PPARs during mycobacterial infections, and Mycobacterial-induced PPAR-mediated autophagy. I believe the authors did a great job, but something needs to be modified:
Response: We thank the reviewer for the comments. We have amended the text to address the reviewer’s concerns.
Point 1: L100: "M. leprae cell wall lipids also contain an LCFA called mycolic acid." I don't think that M. leprae cell wall contains only one kind of mycolic acid. According to "Quantitative Comparison of the Mycolic and Fatty Acid Compositions of Mycobacterium leprae and Mycobacterium gordonae", M. leprae contains alpha-mycolic acids and ketomycolic acids.
Response 1: We thank the reviewer for this comment to improve our manuscript. We have added the reference and revised the sentence as follows (page 3, lines 100-101):
“M. leprae cell wall lipids also contain mycolic acids, other types of LCFAs typical for mycobacteria, such as alpha-mycolic acids and ketomycolic acids [Minnikin, 1985 #170].”
Reviewer 2 Report
Manuscript is a review of recent findings that describe the activation of PPARs by mycobacterial infections and their role in determining the fate of bacilli by inducing lipid metabolism, anti-inflammatory function, and autophagy. The manuscript is well written and the data are sounds. References are pertinent.
No major changes are required.
Author Response
Response: We are grateful for the peer review.